# A Low-Cost On-Street Parking Management System Based on Bluetooth Beacons [note 1]

**DOI:** 10.3390/s20164559

**Published:** 2020-08-14

**Authors:** Chi-Fang Chien, Hui-Tzu Chen, Chi-Yi Lin

**Affiliations:** Department of Computer Science and Information Engineering, Tamkang University, New Taipei City 25137, Taiwan; 607410320@s07.tku.edu.tw (C.-F.C.); pkdog82@gmail.com (H.-T.C.)

**Keywords:** Bluetooth beacons, Kalman filter, received signal strength indicator, smart parking

## Abstract

In recent years, many city governments around the world have begun to use information and communication technology to increase the management efficiency of on-street parking. Among various experimental smart parking projects, deployment of wireless magnetic sensors and smart parking meters are quite common. However, using wireless magnetic sensors can only detect the occupancy of parking spaces without the knowledge of who are currently using these parking spaces; human labor is still needed to issue the parking bills. In contrast, smart parking meters based on image recognition can detect the occupancy of parking spaces along with the license plate numbers, but the cost of deploying smart parking meters is relatively high. In this research, we investigate the feasibility of building an on-street parking management system mainly based on low-cost Bluetooth beacons. Specifically, beacon transmitters are installed in the vehicles, and beacon receivers are deployed along the roadside parking spaces. By processing the received beacon signals using Kalman filter, our system can detect the occupancy of parking spaces as well as the identification of the vehicles. Although distance estimation using the received signal strength is not accurate, our experiments show that it suffices for correct detection of parking occupancy.

## 1. Introduction

On-street parking management has long been a topic of concern for many crowded cities around the globe. According to [1], free parking imposes non-negligible economic and environmental costs on the cities. To this end, traditionally municipalities install parking meters to collect money or hire workers to issue parking bills. However, this does not solve the whole problem. Drivers still need to drive around the streets looking for a vacant parking space and their vehicles cause traffic congestion. Thanks to the recent advancement of information and communication technology (ICT), various smart parking systems based on sensors and wireless networks are proposed. As pointed out in a white paper [2] published by the GSM Association, the benefits of smart parking technologies include reduced traffic congestion, reduced air pollution, reduced time taken to find a parking space, and increased revenue per parking space.

The SFpark pilot program [3] implemented by the San Francisco Municipal Transportation Agency (SFMTA) is one of the earliest smart parking initiatives in the world, which employs wireless magnetometers to detect real-time parking space occupancy in metered spaces. After drivers park their cars, they pay at the smart parking meters for their intended duration of stay. A unique feature of the program is the demand-responsive parking pricing policy. Their experimental results showed that it becomes easier for drivers to find a parking spot and the hourly cost of parking for the drivers actually goes down. The City of Melbourne takes a similar approach as in the SFpark program: sensors are embedded into the asphalt to detect when a vehicle has entered the parking space, and the driver then pays the fee for the allocated time at a smart meter or with the PayStay smartphone app [4]. The on-street parking data from the in-ground sensors is made available for public use through an open data platform [5], which indicates the availability of the parking spaces, the allowed length of time for each parking space, and whether a parking space is restricted to people with a disabled parking permit.

In Taiwan, a common practice used by all the on-street parking management authorities is that parking attendants must patrol along the streets to issue parking bills. Parking attendants also need to circle around the streets every 30 min because the unit of charging is half an hour. Since 2017 many city governments have begun the trials of smart on-street parking systems. In the trials conducted by Taipei City, two types of sensors are deployed: wireless magnetometers, and image recognition-based smart parking meters [6]. When a vehicle enters a specific parking space and the change of occupancy is detected by the wireless magnetometer, the sensor sends the event to the parking database through the NB-IoT interface. Then, a parking attendant will be notified and dispatched to this parking space to issue a parking bill. For those parking spaces installed with smart parking meters, there is no need to dispatch a parking attendant to issue a parking bill because the license plate number can be detected and billed accordingly. Drivers may pay by smart cards at the smart parking meters, or use the ParkingLotApp smartphone app [7] to pay on-line. Although smart parking meters remove the burden of human labor almost completely, their high cost (about 2000~4000 USD per unit) hinders them from being deployed at the whole Taipei City with more than 40,000 on-street parking spaces. Furthermore, it is possible that smart parking meters may fail to acquire the identity of vehicles under poor illumination conditions or obstruction of license plates. Magnetometers are less expensive (about 100~200 USD per unit) and do not suffer from illumination or obstruction problems, however they cannot acquire the license plate numbers to bill the drivers automatically. This motivates us to develop a low-cost alternative solution to achieve the goals of detecting the occupancy of on-street parking spaces as well as identifying the identity of vehicles currently using these parking spaces. Based on the system architecture proposed in our previous work [8], here we incorporate the technique of Kalman Filter and develop a more complete system.

Our approach is briefly described as follows. We assume that vehicles are equipped with a Bluetooth beacon transmitter, and the on-street parking spaces are installed with Bluetooth beacon receivers. Since the Bluetooth beacon technology is based on Bluetooth Low Energy (BLE) physical interfaces, beacon transmitters can run on coin batteries and last for several years with a long duty cycle. When a vehicle enters a parking space, its beacon packets containing a specific identifier will be detected by the beacon receivers. At the beacon receivers, the Received Signal Strength Indication (RSSI) of the beacon packets will be processed using the Kalman Filter and then sent to the gateway. By comparing the measurements from the beacon receivers, the gateway is able to determine the occupancy of the parking space used by this vehicle, and finally sends the information to the parking management system to count its duration of stay. When the vehicle leaves the parking space, its beacon packets are no longer detectable by the beacon receivers. Accordingly, the parking management system may calculate the total duration of stay of the vehicle and then issue an electronic parking bill to the vehicle owner.

Overall, this paper makes the following contributions:(1)The cost of our on-street parking management system is a lot more inexpensive than existing systems relying on either wireless magnetometers or image recognition-based smart parking meters.(2)Compared with magnetometer-based approaches that only detect the occupancy of the parking spaces, our approach can detect the occupancy and acquire the identity of vehicles using the parking spaces at the same time.(3)Compared with image recognition-based approaches that may fail under poor illumination conditions or obstruction of license plates, our approach can correctly detect the identity of vehicles using the parking spaces through Bluetooth radio frequency signals.(4)We identify some limitations of our Bluetooth beacon-based approach and propose possible solutions to cope with them.

The remainder of this paper is organized as follows: In Section 2 we will discuss the related work and the technology background. In Section 3 we will describe the system architecture and explain the principle of operations. Our implementation details, experimental results, and some discussions are described in Section 4. Finally, Section 5 concludes our work and gives future directions.

## 2. Related Work

In this section, we mainly review the existing smart parking systems in the literature, and give a brief introduction to the Bluetooth beacon technology.

### 2.1. Review on Smart Parking Systems

Zhang et al. [9] developed a street parking system (SPS) based on wireless sensor networks. They deployed magnetic sensor nodes on every parking space and designed a vehicle detection algorithm. When a sensor detects a vehicle entering or leaving the parking space, it transmits a message to the base station through the ZigBee wireless interface. The main contribution of their work is the adaptive sampling mechanism running in the sensor nodes which balances the energy consumption and the accuracy of detection. Barone et al. [10] proposed an intelligent parking assistant (IPA) architecture where inductive magnetic loops and RFID readers are deployed on every parking space to detect the occupancy state of the space and the RFID tag associated with the driver, respectively. They also allow drivers to reserve a parking space at a specific destination area before the driver departs for the destination.

Some researches focus on detecting the available on-street parking spaces without the need to deploy infrastructure sensors. For instance, both the ParkNet system developed by Mathur et al. [11] and the work by Roman et al. [12] mounted ultrasonic sensors on the side of a probe vehicle (e.g., taxis, buses) to measure the distance from the vehicle to the roadside. The probe vehicles are also with a GPS receiver so that the collected distance data can be associated with GPS coordinates. In [12] a supervised learning algorithm was developed to recognize parked cars and empty spaces. To handle the inaccuracy in the received GPS location information, [11] and [12] use an environmental fingerprinting approach and a new map matching technique, respectively. Unfortunately, the successful detection rate of both approaches depends heavily on the quality of the GPS readings, and both require a more sophisticated mechanism in multi-lane scenarios where lane-changing behaviors must be considered.

Apart from using probe vehicles to detect available on-street parking spaces, some researches such as [13,14,15] favor the mobile crowd sensing (MCS) model which leverages the power and wisdom of the crowd. In MagnoPark [13] proposed by Arab et al., the smartphones of pedestrians on the sidewalks serve as mobile sensors, which collect accelerometer, gyroscope, magnetometer, and GPS data. The main data used by their system for occupancy classification is the magnetometer readings, which may change when a vehicle is nearby the smartphone. In ParkCar [14] proposed by Banti et al., the tasks of identifying a vacant parking space are entirely executed by humans (called participants) in response to the drivers (called requestors) who issue requests to find one in the areas of interest. To reward the participants for their contribution, the authors designed a rule that considers (1) the number of returned answers, (2) the distance of the returned spot to the point of interest (POI), and (3) the time taken to submit the answer, in the form of virtual currency that may be used to gain access to other services. In ParkCrowd [15] proposed by Shi et al., crowd workers use a smartphone app to report real-time parking availability information to a cloud server. The cloud server aggregates the collected data and employs a joint probabilistic estimator to infer the future availability of parking spaces based on crowdsensed knowledge. They also presented the idea of evaluating the knowledge of crowd workers based on the answers collected from the workers for the POI questions, so as to improve the reliability of the information being disseminated.

Seymer et al. [16,17] proposed a Bluetooth beacon-based smart parking solution for off-street parking lots, which aims at eliminating ticketing and management infrastructure. As in our approach, the authors assume that vehicles using the parking lot are equipped with Bluetooth beacon transmitters. However, in the parking lot only a few beacon receivers are installed. That is, per-space occupancy detection is not needed. These beacon receivers not only detect the beacon packets, but also form a mesh network to forward the observations to the central node for processing. To locate the parked vehicles, they take the radio fingerprinting approach by using the RSSI from the Bluetooth beacon transmitters, and then use the random forest machine learning model to predict the parked locations based on the fingerprints.

In [18], Mackey et al. designed a smart parking system based on Bluetooth beacons, where beacon transmitters are placed at the side of each parking spot location, and the smartphones of the drivers serve as beacon receivers. When a driver parks the car in a specific parking space, the driver’s smartphone may detect the nearest beacon transmitter. By acquiring the unique identifier in the beacon packets from this nearest beacon transmitter, the driver can register the vehicle with the corresponding parking space through a mobile application. Specifically, the authors use Google’s Eddystone beacon format to carry an encoded uniform resource locator (URL) for the drivers to register their vehicles. When the vehicle is about to leave the parking space, the driver unregisters from the parking space and then payment is automatically processed based on the duration of stay. To increase the accuracy of determining the nearest beacon transmitter, the authors implemented the particle filtering algorithm in the parking management server to deal with the RSSI measurements. Experimental results show that using particle filtering performs slightly better than using the raw data in distance estimation.

### 2.2. Overview of Bluetooth Beacon Technology

Bluetooth beacon packets are transmitted over the three advertising channels, carried in BLE advertising packet data units (PDU). In Bluetooth version 4.1 [19], the maximum size of a BLE advertising channel PDU is 39 bytes, in which the advertising data field is up to 31 bytes. Although in Bluetooth version 5.0 the maximum advertising PDU size has been extended to 257 bytes, the legacy advertising PDU of up to 39-byte long is preserved in the specification in order to maintain the backward compatibility. There are several Bluetooth beacon pseudo-standards in the industry such as Apple’s iBeacon [20], Google’s Eddystone [21], and Radius Networks’ AltBeacon [22]. Figure 1 shows their packet formats, respectively. Figure 1a is the iBeacon packet format, in which the following four fields are notable: 16-byte Universally Unique Identifier (UUID), 2-byte Major, 2-byte Minor, and 1-byte Tx Power. According to the white paper [23], a UUID should be specific to an application and deployment use case, and the Major/Minor values are used to hierarchically specify sub-regions within a larger region or use case. As for Tx Power, it is defined as the wireless signal strength at a distance of 1 m away from the beacon transmitter. By comparing the RSSI with the value of Tx Power embedded in the received beacon packet, a beacon receiver can estimate its proximity to the beacon transmitter. The packet formats of Eddystone-UID and AltBeacon are shown in Figure 1b,c, respectively. Note that Eddystone actually defines four frame types: Eddystone-UID, Eddystone-URL, Eddystone-TLM, and Eddystone-EID. Since our work focus on identifying parked vehicles, we only introduce the most relevant frame type: Eddystone-UID. In Eddystone-UID the 10-byte Namespace field and the 6-byte Instance field can be combined to form a 16-byte universally unique beacon ID. The field of Ranging Data is designed to carry the calibrated transmitter power at 0 m, which can be obtained by measuring the actual signal strength from 1 m away and then add 41 dBm to the value. As for AltBeacon, use of the 20-byte Beacon ID is indicated by the specification that the first 16 bytes of the Beacon ID should be unique to the advertiser’s organizational unit for interoperability purposes. The 1-byte Ref RSSI field is used to represent the average received signal strength at 1 m from the advertiser, which is exactly the same as the definition of the Tx Power in iBeacon packets. To conclude, these beacon packets serve a common purpose: carrying a unique identifier as the identification of the transmitter, along with a transmitter power value for proximity estimation at the receiver.

## 3. The Proposed Parking Management System

In this section, we will describe the proposed on-street parking management system based on Bluetooth beacons. We start from presenting the system architecture, followed by the formation of the parking sensor network, then discuss the principle of operations of the system.

### 3.1. System Architecture

Figure 2 shows the architecture of our on-street parking management system. There are four roles in the system: beacon transmitter, beacon reader, gateway, and parking server. In the following we give detailed descriptions on the four roles, and their relationships are shown in Figure 3.
(1)*Beacon transmitter*: Drivers eligible to use the smart on-street parking service are required to register their vehicles in order to get a registered beacon transmitter. We assume that the beacon transmitter is attached to the right-side mirror of the vehicle (for right-hand traffic countries), which is closer to the curb when the vehicle is parked. The beacon transmitter broadcasts beacon packets regularly, which then can be detected by the beacon readers. The source Bluetooth MAC address of the beacon packets from a specific vehicle is fixed, therefore the beacon readers are able to distinguish the owners (i.e., vehicles) of the received beacon packets.(2)*Beacon reader*: As shown in Figure 2, beacon readers are deployed at the corners of the parking spaces along the curb. Readers periodically scan for beacon packets with the service-specific UUID from the registered beacon transmitters. Once the UUID is matched, the reader will process the packet’s RSSI using Kalman filter and then generate a distance estimation. Finally, the reader publishes the following data to the gateway: the detected Bluetooth MAC address, the estimated distance, and the time of detection.(3)*Gateway*: The main functionality of the gateway is to determine the occupancy state of each parking space based on the data provided by the beacon readers. Specifically, by comparing the estimated distances of the same beacon transmitter measured at various readers, the gateway is able to infer that the vehicle has been parked at a specific parking space. Then, the gateway sends the findings to the database in the remote parking server.4)*Parking server*: The parking server is responsible for storing the occupancy state of all the on-street parking spaces along with the identity of the vehicles using the parking spaces. The administrator can log on to the system and then query the database to view the real-time occupancy information as well as the parking history records. With the information, the management system would be able to generate electronic parking bills for the drivers. On the other hand, the parking server can also be used by the drivers to view their own parking records.

### 3.2. Formation of the Parking Sensor Network

As shown in Figure 2, beacon readers deployed along the curb and a nearby gateway form a parking sensor network. The proximity information detected from the beacon transmitters attached to the parked vehicles must be sent back to the gateway in order to determine the occupancy state of the parking spaces. Since we focus on offering a low-cost infrastructure-based solution, the sensor devices used in our system are assumed to be low-power and resource-constrained. At the same time, we would like the parking sensor network capable of running the IP protocol to ease the management of the sensor nodes as well as deploying application-layer protocols for future development. Consequently, we choose to organize the parking sensor network based on IPv6 over Low-power Wireless Personal Area Network (6LoWPAN) technology.

According to IETF RFC 7668 [24], the Bluetooth network for IPv6 networking purposes follows a star topology. In our system, the gateway playing the Bluetooth central role maintains a separate connection to each beacon reader which plays the Bluetooth peripheral role. Since the gateway needs to communicate with the parking server in the public Internet, it serves as the 6LoWPAN Border Router (called 6LBR) which interfaces the 6LoWPAN network with the Internet, while all beacon readers serve as the 6LoWPAN Nodes (called 6LN). The procedure of setting up the 6LoWPAN network follows the three steps:(1)Establish a link-layer connection between the 6LBR and each 6LN.(2)Establish a Logical Link Control and Adaptation Protocol (L2CAP) channel on top of the link-layer connection.(3)Configure IPv6 global address for the Bluetooth interfaces.

### 3.3. Principle of Operations

In our implementation, we choose the Apple iBeacon format for the beacon packets. As described in Section 3.1, we use a specific UUID for our parking service which needs to be configured in the beacon transmitters. This service-specific UUID serves as the filtering criterion for the beacon readers to exclude irrelevant beacon packets from been processed. In the following we use the scenario in Figure 2 as an example to describe the principle of operations in our system.

Assume that a vehicle enters the parking space 02 at time *t*_1_. By scanning at a fixed interval (say 10 sec), the neighboring beacon readers will discover the beacon packets associated with this vehicle very shortly, and then insert the MAC address of the beacon transmitter (assuming 11:22:33:44:55:66) and the timestamp (shortly after *t*_1_) into the local database. Meanwhile, the RSSI value of the detected beacon packet will be used to initialize the Kalman filter. During the subsequent scanning intervals, the beacon packets from the same vehicle will be detected repeatedly. With a sequence of RSSI measurements from the same beacon transmitter, each beacon receiver can perform Kalman filtering to get a smoother RSSI reading. Use of Kalman filter to smooth the RSSI measurements of Bluetooth beacon signals is a quite common technique in the literature, such as [25,26,27], just to name a few. Since that a one-dimensional Kalman filter suffices for RSSI filtering, its low computational complexity makes it feasible to be implemented in resource-constrained IoT devices such as Raspberry Pi 3, and the filtered RSSI value can be derived almost immediately after a new RSSI measurement arrives.

The next step is to use the filtered RSSI value and the Tx Power value in the beacon packet to give an estimated distance between the transmitter and the receiver. We adopted the equation used by Android Beacon Library [28] for estimating the proximity of the beacon transmitter from the beacon receiver as follows:
*d* = 0.89976 × (RSSI/*A*)^7.7095^ + 0.111(1)
where *d* is the estimated distance and *A* is the Tx Power value at the distance of 1 m. After the estimated distance is determined, each reader sends its own data as a list of four-tuples (reader ID, MAC address of the transmitter, estimated distance, time of detection) to the gateway. In our implementation, we choose Message Queuing Telemetry Transport (MQTT) [29], the most popular IoT protocol for publish/subscribe messaging, as the communication protocol between the beacon readers and the gateway. Specifically, beacon readers act as the publishers who push their data to the gateway acting as the broker. Since the gateway is also responsible for determining the occupancy state of the parking spaces, it is assigned the role of the subscriber at the same time in order to acquire the data and process them accordingly.

Assume that the gateway successfully acquired the four pieces of data shown in Table 1. Note that the time of detection of the four pieces of data may be slightly different because the scanning periods of the readers need not to be aligned. To determine the parking space occupied by this vehicle, the gateway can simply sort the tuples by the estimated distances, which indicates that the vehicle is closest to Reader 2, followed by Reader 3. Meanwhile, the gateway further found that the smallest distance is below a predetermined threshold (e.g., 2 m). Finally, the gateway can conclude that the vehicle is on the parking space between Reader 2 and Reader 3, which is parking space 02. Once the occupancy information is determined, the gateway stores the following 5-tuple (MAC address of the transmitter, start time, end time, parking duration, parking space ID) locally, and sends a copy to the database in the parking server, where start time refers to the time of the vehicle’s first appearance recorded locally, end time is the time at which the occupancy information is determined, and parking duration is end time minus start time. Note that the value of end time will be updated by the following rounds of determining the occupancy information, as long as the vehicle stays on the same parking space.

When the vehicle leaves the parking space at time *t*_2_, the readers no longer receive the beacon packets associated with this vehicle after *t*_2_. During the subsequent scanning intervals, no record with the MAC address 11:22:33:44:55:66 would ever appear in any reader. If the number of times without sensing this vehicle exceeds a certain threshold, the readers will delete the record associated with this vehicle. Likewise, the gateway also counts the number of times without receiving the record of this vehicle from the readers. When the number exceeds the threshold, the data associated with this vehicle will be deleted from the gateway, and a notification will be sent to the parking server. Accordingly, the parking server moves the record associated with this vehicle to the *parking history* table, and uses the final parking duration of the vehicle (roughly *t*_2_–*t*_1_) to issue an electronic parking bill to the vehicle owner.

## 4. Preliminary Results and Experiments

In this section, we will show our preliminary results on the prototype implementation. Specifically, we introduce the web interfaces and an iOS app for our parking management system which are designed for both the drivers and the administrative staffs, and the data displayed on the interfaces are real data. Furthermore, we will show the experiments conducted to verify the effectiveness of the proposed system and give some discussions.

### 4.1. Parking Management System

Our parking server also serves as a web server to allow drivers and administrative staffs to access the smart parking service. The web server is based on the open-source WordPress software, which supports responsive design. Therefore, drivers and administrative staffs are freely to access the parking management system with desktop browsers or mobile browsers. The main interface of our parking management system is shown in Figure 4a,b which are the web interface and the iOS app interface, respectively, and both display the content of the Parking Space Overview page. 

Note that in our prototype we implemented the system architecture as in Figure 2 for experimental purposes, so there are only three parking spaces totally. From this page drivers and/or passengers are able to check the real-time availability of these parking spaces before they arrive at the location. If drivers would like to sign up for the smart parking service, they can register their vehicles through the Vehicle Registration page shown in Figure 5a. Specifically, drivers enter the beacon MAC address, driver ID, and the license plate number to register.

As for the Parking Record and Payment page, drivers may enter the combination of their own driver ID and the license plate number to lookup their parking records and then pay the parking bills. As shown in Figure 5b, there are two parking records associated with license plate number 6666-CHI, for each record the parking details including parking space ID, start time, end time, and parking duration are shown. The Pay button below can be used for the drivers to pay the parking bills online. On the same page, administrative staffs may enter a secret combination of driver ID and license plate number as a pass phrase to lookup the detailed real-time state of all the parking spaces, as the screenshot shown in Figure 6.

### 4.2. Experiments

As stated in Section 3, here we assume right-hand traffic countries, so the beacon transmitters are attached to the right-side mirror of the vehicles. According to the traffic regulation, vehicles using the on-street parking spaces should be parked facing forward toward the direction of traffic. As a consequence, the right side of the vehicles parked on two-way streets must be closer to the curb. However, on one-way streets, if vehicles are parked on the left side of the road, the left side of the vehicles will be closer to the curb. Although uncommon, some careless drivers may park their vehicles facing backward toward the direction of traffic. To validate the usability of the proposed system, we conducted experiments covering all the three conditions. The beacon receivers and the gateway are Raspberry Pi 3 development boards running Raspberry Pi OS, and the beacon transmitters are Estimote Proximity Beacon devices. The list of equipment is shown in Table 2. The experiments were conducted in the open space of our Main Engineering Building, shown in Figure 7. The scanning interval of the beacon readers is 10 sec, and the duration of each experiment is about 10 min.

#### 4.2.1. Experiment 1

In this experiment, the test scenario is shown in Figure 8a where we emulate two vehicles parking on parking space 01 and 02, respectively, with their right side closer to the curb. Considering the average height of the side mirrors and the sidewalk, we deploy the transmitter and the receiver 1.1 m and 0.15 m above the ground, respectively. The detailed deployment within each parking space is shown in Figure 8b.

Figure 9a shows the estimated distances of transmitter 1 from the four readers. It is clear that even been processed by the Kalman Filter, the estimated distances are still very unstable, and their values are far from the actual distances. 

For example, although reader 1 is the nearest reader to transmitter 1, in most of the detection rounds reader 2 is been estimated as the nearest to the transmitter. To determine the parking space used by the vehicle, we sort the estimated distances produced by the four readers in ascending order in each detection round and show the reader IDs of the top-2 smallest observations in Figure 9b. During the first 4 min, the top-2 reader IDs alternated between (1, 2) and (2, 3) frequently, but finally stabilized and converged to (1, 2) which corresponds to parking space 01.

Figure 10a shows the estimated distances of transmitter 2 from the four readers. Again, we sort the estimated distances produced by the four readers in ascending order in each detection round and show the reader IDs of the top-2 smallest observations in Figure 10b. Until about 3 min, reader 2 finally been recognized as the top-1, but the second smallest distance alternated between reader 1 and reader 3 for a while. This causes oscillation of the detection result between parking space 01 and 02. At about 9 min, it finally stabilized and converged to (2, 3) which corresponds to parking space 02.

#### 4.2.2. Experiment 2

In this experiment, the test scenario is shown in Figure 11a where we emulate two vehicles parking on parking space 01 and 02, respectively, with their left side closer to the curb. Again, we deploy the transmitter and the receiver 1.1 m and 0.15 m above the ground, respectively, as shown in the detailed deployment in Figure 11b.

The experimental results are shown Figure 12, where we skip the estimated distances from the four readers and show only the process of determining the occupancy for vehicle 1 and vehicle 2. 

From Figure 12a we can see that from the beginning, reader 2 has been recognized as the nearest reader to transmitter 1, and the detection result is correct. However, the estimated distances produced by reader 1 and reader 3 are very close to each other, causing the detection result alternated between parking space 01 and 02 for a while. Starting from 5 min, the second nearest reader ID is stabilized, which indicates that the vehicle is on parking space 01. Figure 12b shows the process of determining the occupancy for vehicle 2. Reader 3 is the nearest one to transmitter 2, which has been successfully detected after about 2 min. We can also see that reader 2 is successfully recognized as the second nearest reader after about 3 min, which indicates that vehicle 2 is on parking space 02.

#### 4.2.3. Experiment 3

As stated in the beginning of this section, careless drivers may sometimes park their vehicles facing backward toward the direction of traffic. Consequently, the scenario in this experiment may happen: two vehicles parking on parking space 01 and 02, respectively, but vehicle 1 is facing backward as shown in Figure 13a. Please also refer to Figure 13b showing the detailed deployment of the transmitters and the receivers in this test scenario, where we can see that reader 2 is the nearest reader to both transmitter 1 and transmitter 2.

Figure 14a shows the process of determining the occupancy for transmitter 1. Just like the case of vehicle 1 in experiment 2, here the nearest reader to transmitter 1 is reader 2, but during the first 5 min the estimated nearest reader alternates between reader 1 and reader 2. Nevertheless, the top-2 smallest observations come from reader 1 and reader 2 all the time. Therefore, from Figure 14a we can see that vehicle 1 is recognized as using parking space 01 from the beginning of the experiment. Figure 14b shows the progress of determining the location of vehicle 2. We can see that although reader 3 is the second nearest to transmitter 2, in the beginning reader 1 is recognized as the second nearest one due to the inaccuracy of the RSSI measurements. After about 5 min, the estimated distance from reader 3 finally stabilizes and becomes smaller than that of reader 1, which indicates that vehicle 2 is on parking space 02.

### 4.3. Discussions

In this section, we analyze the errors in distance estimation observed from the experiments, as well as some discussions on the limitations of our current system. Finally, we make a detailed comparison between the existing infrastructure-based on-street parking systems and our proposed system.

Errors in distance estimation

To evaluate the accuracy of distance estimation using RSSI, we collect all the estimation results from the three experiments in Section 4.2 and summarize them into the boxplots shown in Figure 15. According to the deployment of beacon transmitters and beacon receivers in the experiments, there are 10 actual transmitter-to-receiver distances. The boxplots of the estimated results at the 10 distances are shown with different colors. Figure 15a,b display the distribution of estimated distances with data from the first 5-min duration and the second 5-min duration, respectively. We can see that both show significant error distances, and the errors of the second 5 min are even larger than those of the first 5 min. However, the estimates of the first 5 min are more unstable, and the estimated distances at the actual distance of 6.58 m resemble the behavior at the actual distance of 1.80 m, which makes it difficult to correctly determine the parking space occupancy for a vehicle. By contrast, the estimates of the second 5 min are more stable, and the estimation results within 6.5 m basically follow the trend of the actual distances. Since the determination of parking space occupancy for a vehicle is based on the top-2 smallest observations with actual transmitter-to-receiver distances smaller than 5 m, from the analysis we can fairly conclude that our system is able to correctly determine the occupancy of vehicles within 10 min given the scanning interval of 10 sec. Note that if we reduce the scanning interval, the time taken to correctly determine the occupancy of vehicles can also be reduced.

Regarding the significant estimation errors shown in our experiments, a similar behavior can be found in [18] that even with computationally expensive filters like particle filter, there exists significant errors in distance estimation using the RSSI of Bluetooth beacons. Nevertheless, compared with [18] we use a more lightweight one-dimensional Kalman filter which is more suitable to be implemented on resource-constraint roadside devices such as Raspberry Pi 3 development boards. Although distance estimation based on the RSSI is inaccurate, we follow the suggestion given by [30]: “*avoid using the absolute value of the RSSI—use the trend instead”*. Specifically, we use the trends of the RSSI perceived by the beacon receivers and compare them to determine the parking space used by a specific vehicle. From our experiments, on average it takes about 5 min for the trend of RSSI to be stabilized, which is equal to about 30 rounds of detection. The delay time won’t be a problem since the final outcome (i.e., the determined parking space ID) will overwrite the previous outcome. Moreover, in many cities, drivers are encouraged to use the on-street parking spaces for their stopping by to reduce the occurrences of double parking. The incentive for the drivers is that if their parking periods are less than 10 min, they can use the parking spaces for free. Although the determination of occupancy for the vehicles may not be accurate during the first 5 min, no parking bills would ever to be issued in this case.

We have also noticed the oscillation of the detection result between adjacent parking spaces such as the case of vehicle 2 in experiment 1 during 6 to 9 min. Again, this is due to the inaccuracy of the RSSI measurements. One possible solution is to adjust the coefficients in Equation (1) for estimating the proximity of the beacon transmitter from the beacon receiver by field tests. However, the environment of the on-street parking spaces across the city may not be uniform, which makes it impractical to use the same coefficients for all the deployed beacon readers. One other possible solution is by human correction. Specifically, when a vehicle enters a specific parking space, assume that the driver’s smartphone can be prompted with a notification showing the parking space ID been detected. If the detected parking space ID is incorrect, the driver can provide the correct ID through the smartphone app.

Detection of vehicles without beacon transmitters

If a vehicle without a beacon transmitter uses the on-street parking space, currently our system would have no way of detecting the occupancy state and the vehicle ID. To solve the problem, we may incorporate wireless magnetometers installed on the parking spaces. The underground wireless magnetometers can detect the existence of vehicles occupying the parking spaces (without knowing the ID of the parked vehicles) and send the occupancy state to the parking server. If a specific parking space is being used by a vehicle without a transmitter, the occupancy state corresponding to this parking space stored in the parking server would show that the space is occupied but the vehicle ID is null. Only when this condition happens, a parking attendant will be dispatched to this parking space to issue a parking bill. In fact, incorporating wireless magnetometers into our system brings another advantage. That is, the problem of oscillating detection results can be solved because the magnetometer can sense the occupancy state directly.

Detection of a vehicle occupying two or more parking spaces

Since we rely on the observed reader IDs of the top-2 smallest transmitter-to-receiver distance estimates to determine the occupied parking space, if one vehicle occupies two or more parking spaces, the current system is unable to detect this situation. If our system is equipped with underground wireless magnetometers, it is still difficult to detect the situation because with a vehicle parking in the middle of two parking spaces, its vehicle body may not necessarily cover the two magnetometers at the same time. To solve the problem, our idea is that the parking app can be added with a “Report For Violation” function for people to report misuses of the parking spaces by uploading a photo taken at the scene. When such a case is reported, a local policeman can be dispatched to the scene to issue a parking ticket.

Misdetection of a passing-by vehicle as occupying a parking space

When vehicles on the street are stopped temporarily waiting for the traffic light to turn green or just passing by the parking spaces, it is possible that their beacon packets happen to be detected by the beacon readers during this short period of time. Without proper handling of the data, the system may misinterpret those vehicles as occupying the on-street parking spaces. To solve the problem, we may add an additional condition that the top-1 smallest distance estimate for a specific beacon transmitter must be lower than a certain threshold to be eligible for occupancy determination. If the top-1 smallest distance estimate is above the threshold, it indicates that the vehicle is not in any of the parking spaces.

Comparison of existing infrastructure-based on-street parking systems

In Section 2.1 we have reviewed some existing smart parking systems. Since our work focuses on lowering the cost of infrastructure-based on-street parking systems, here we compare our work to the existing systems based on magnetometers (e.g., SFpark [3] and City of Melbourne [4]), smart parking meters (e.g., Taipei City [6]), as well as Mackey et al.’s beacon-based parking system [18]. The comparison results are shown in Table 3, where we assume the market prices of a magnetometer, a smart parking meter, a beacon transmitter, and a beacon receiver (e.g., Raspberry Pi Zero) are 100, 2000, 5, and 10 USD, respectively, and the constant *N* refers to the number of on-street parking spaces along a specific side of a street. As pointed out in Table 2, in terms of automatic detection of both parking space occupancy and the vehicle ID, only image recognition-based smart parking meters and our system can achieve that. In terms of the number of roadside devices, since we deploy beacon readers at the corners of the parking spaces along the curb, *N* parking spaces require *N* + 1 beacon readers. For all other systems, they only need one device per parking space. When we multiply the number of roadside devices by the unit price of the devices, it is obvious that Mackey’s beacon-based system and our system are a lot more inexpensive than the other two systems. Although Mackey’s system has a lower cost than our system, their system requires drivers to manually register and un-register to the parking spaces. Furthermore, human labor is still required to patrol along the streets in order to check whether each vehicle has been registered or not. Note that our system also requires each vehicle to install a beacon transmitter. Although this can be a nonnegligible effort, thanks to the low cost of beacon transmitters, the one-time cost of installing beacon transmitters on the vehicles is still a lot more inexpensive than the long-term cost of hiring parking attendants.

## 5. Conclusions and Future Work

In this research, we proposed a low-cost on-street parking management system based on Bluetooth beacons for infrastructure-based smart parking application scenarios. Existing solutions that rely on wireless magnetometers or image recognition-based smart parking meters are either insufficient or too expensive in terms of their high cost of human labor or dedicated hardware. Specifically, compared with magnetometer-based systems which only detect the occupancy of parking spaces, our system can achieve the goal of detecting the occupancy and acquire the identity of vehicles using the parking spaces at the same time, which saves the high labor costs. Compared with image recognition-based smart parking meters, our system does not suffer from the problems of poor illumination or obstruction of license plates, and the hardware cost of our system is much lower than smart parking meters.

We have built a prototype system and conducted three experiments that cover different usage scenarios to validate the effectiveness of our system. The experimental results showed that on average it takes about 5 min for the trend of RSSI to be stabilized with a scanning interval of 10 sec with the help of a lightweight one-dimensional Kalman filter, so that the parking space ID can be determined accordingly. Although the distance estimates are far from accurate, we can still determine the occupancy of parking spaces correctly. Our system is also user-friendly; drivers and administrative staffs are able to access the system with desktop/mobile browsers and an iOS app we developed. Through the web/app interfaces, users can see the real-time availability of parking spaces, sign up for the smart parking service, view their parking records, and pay their parking bills online.

The current parking sensor network is based on the 6LoWPAN technology specified in IETF RFC 7668, which only supports a star topology. This could pose a limitation on the distance between the gateway and the beacon readers. In the future, we will try to upgrade the parking sensor network based on the ongoing work [31] by the 6lo Working Group which extends the 6LoWPAN technology to support IPv6 mesh over BLE. Furthermore, we plan to incorporate wireless magnetometers into our system to improve the accuracy of occupancy detection and allowing vehicles without beacon transmitters to use the smart parking service.

## Figures and Tables

**Figure 1 sensors-20-04559-f001:**
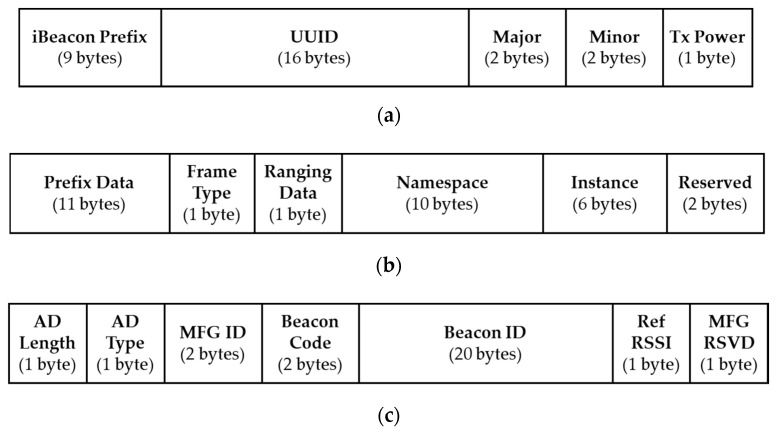
Packet formats of three Bluetooth beacon pseudo-standards: (**a**) iBeacon, (**b**) Eddystone-UID, and (**c**) AltBeacon.

**Figure 2 sensors-20-04559-f002:**
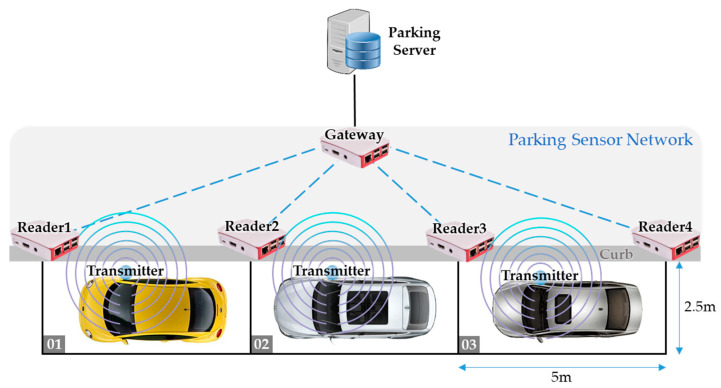
The proposed system architecture.

**Figure 3 sensors-20-04559-f003:**
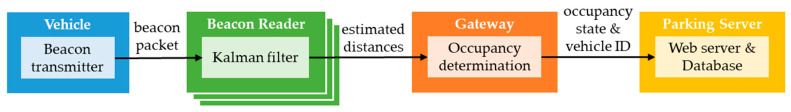
The relationships between the four roles.

**Figure 4 sensors-20-04559-f004:**
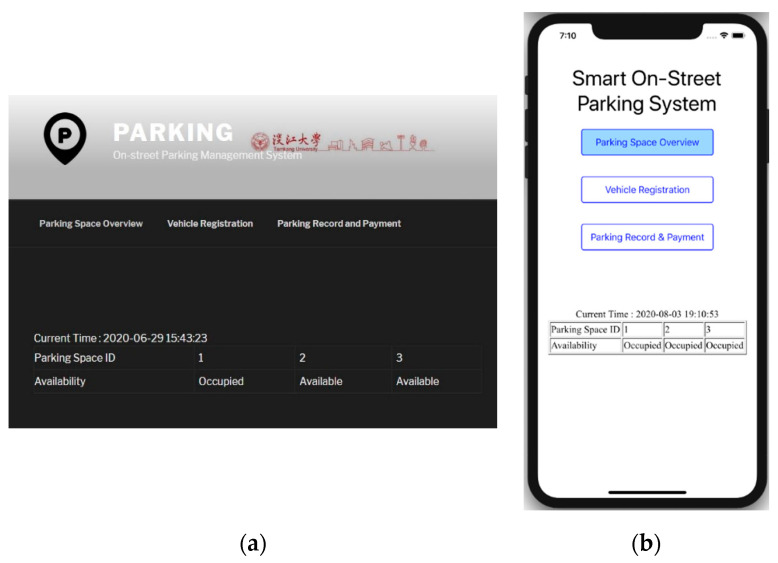
Parking Space Overview page showing availability of the parking spaces: (**a**) desktop web interface (**b**) iOS app.

**Figure 5 sensors-20-04559-f005:**
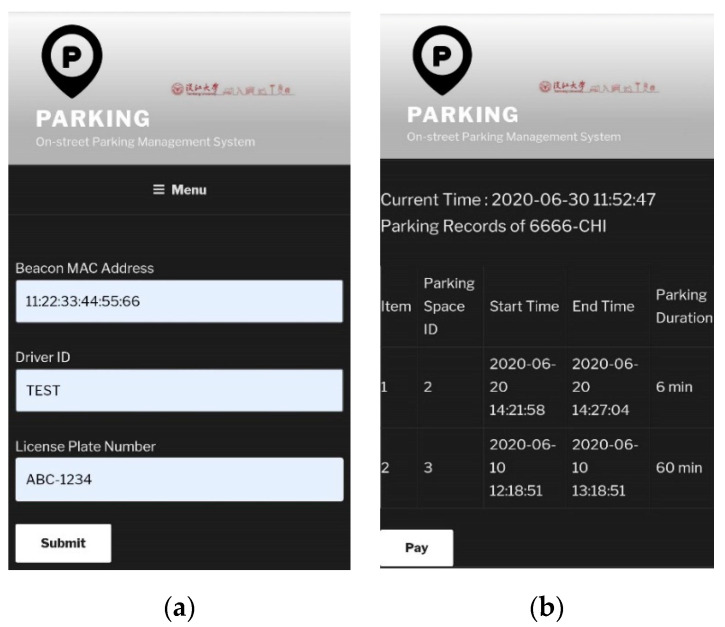
Web interfaces: (**a**) Vehicle Registration page for drivers to fill in registration data, (**b**) Parking Record and Payment page showing the parking records of a specific vehicle.

**Figure 6 sensors-20-04559-f006:**
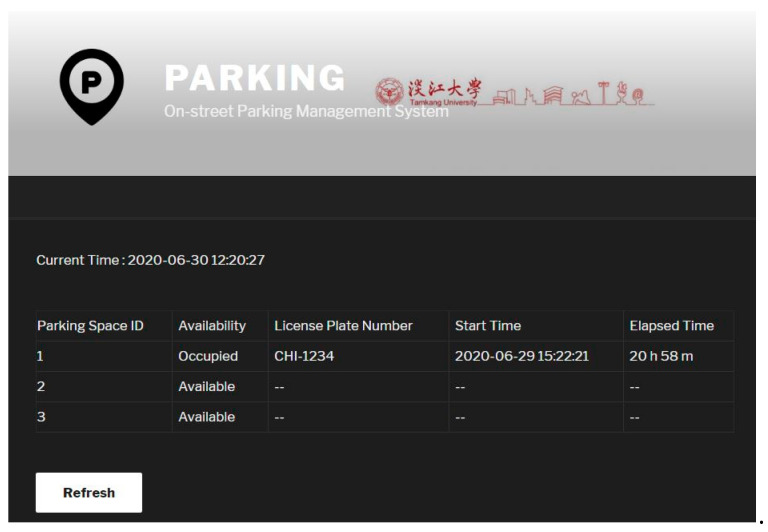
Parking Record and Payment page showing the detailed real-time state of the parking spaces for administrative purpose (desktop version).

**Figure 7 sensors-20-04559-f007:**
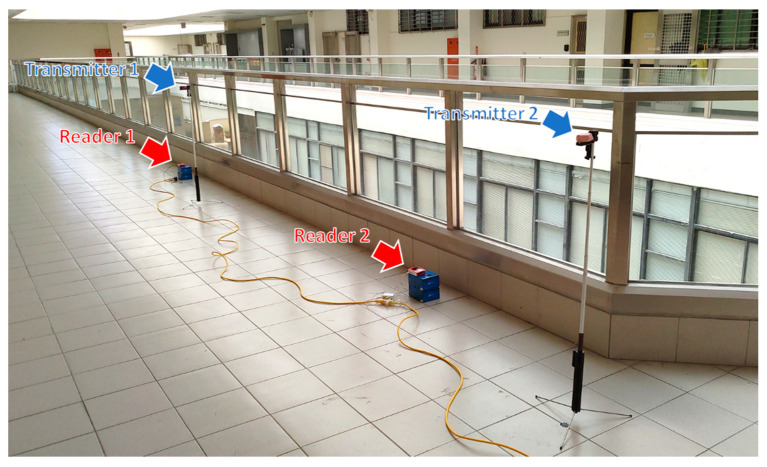
Environment for the experiments.

**Figure 8 sensors-20-04559-f008:**
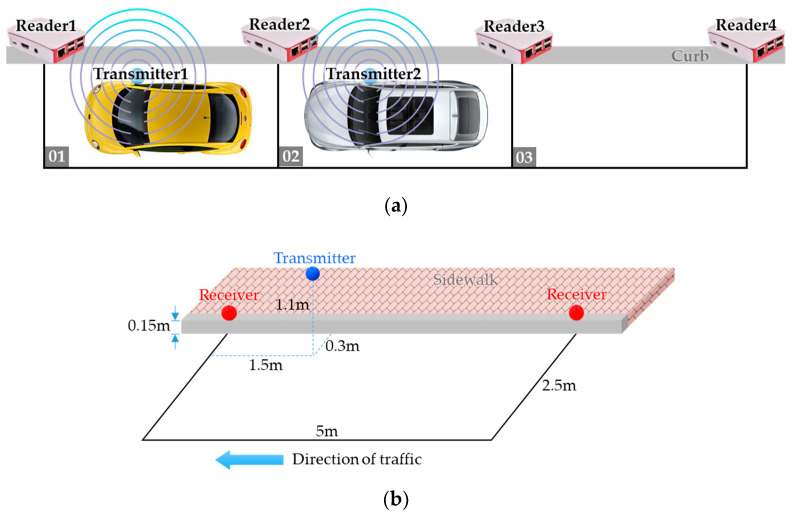
Experiment 1: (**a**) test scenario, (**b**) detailed deployment within each parking space.

**Figure 9 sensors-20-04559-f009:**
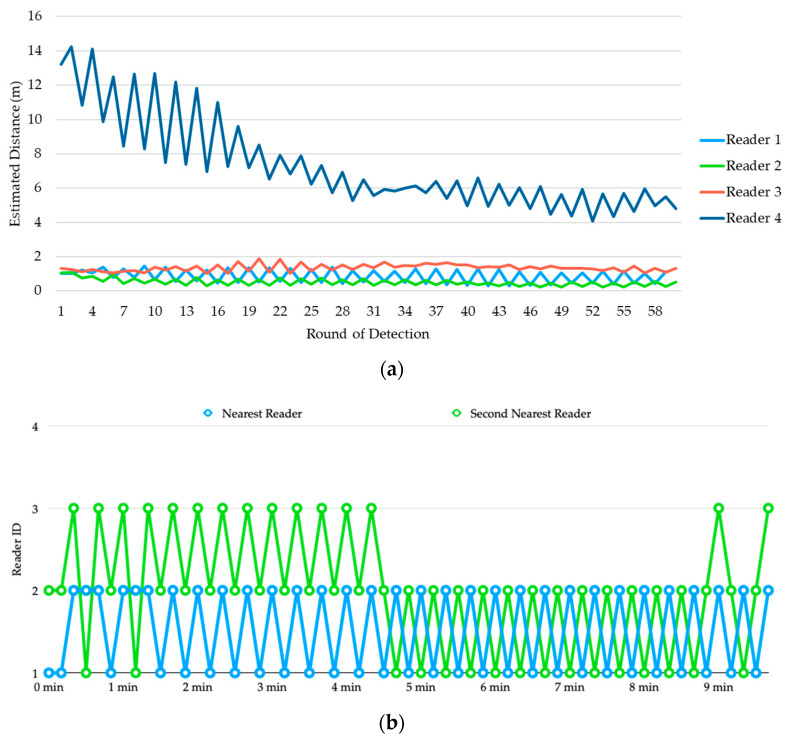
Experiment 1: (**a**) estimated distances of transmitter 1 from the four readers, (**b**) determination of occupancy for vehicle 1.

**Figure 10 sensors-20-04559-f010:**
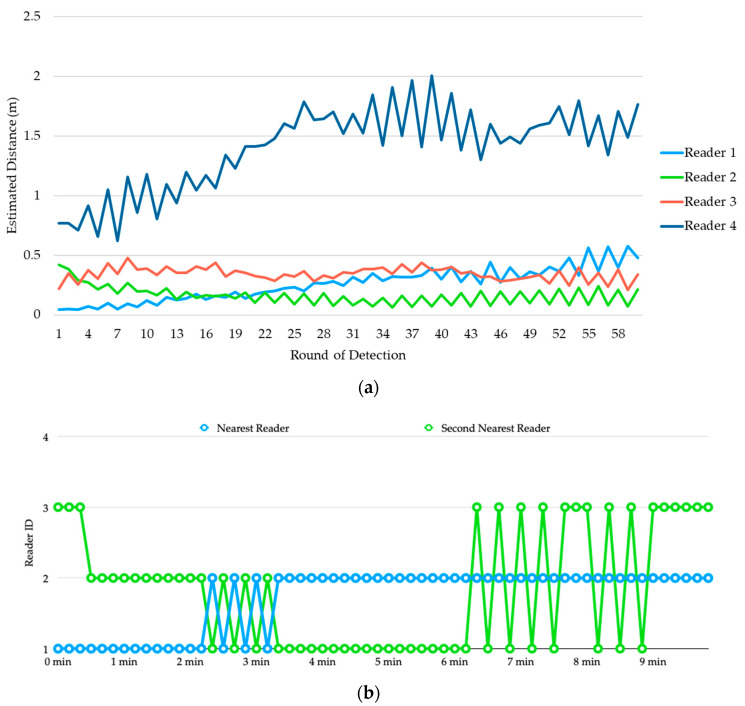
Experiment 1: (**a**) estimated distances of transmitter 2 from the four readers, (**b**) determination of occupancy for vehicle 2.

**Figure 11 sensors-20-04559-f011:**
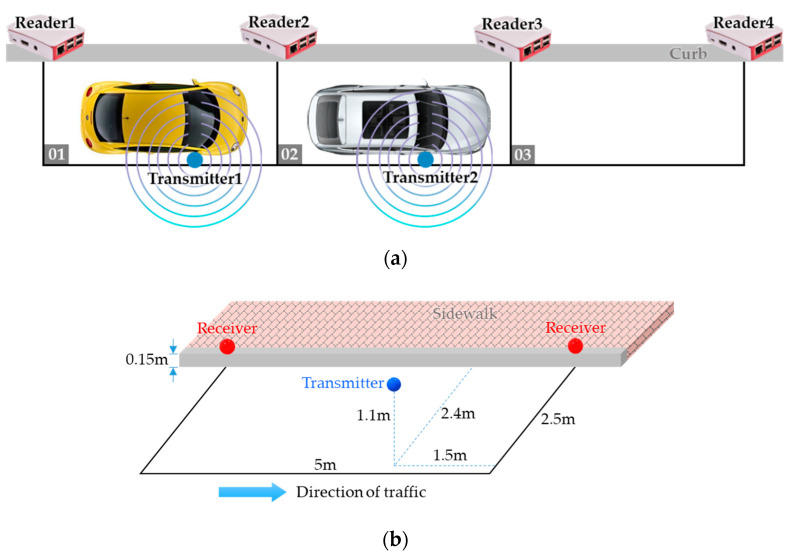
Experiment 2: (**a**) test scenario, (**b**) detailed deployment within each parking space.

**Figure 12 sensors-20-04559-f012:**
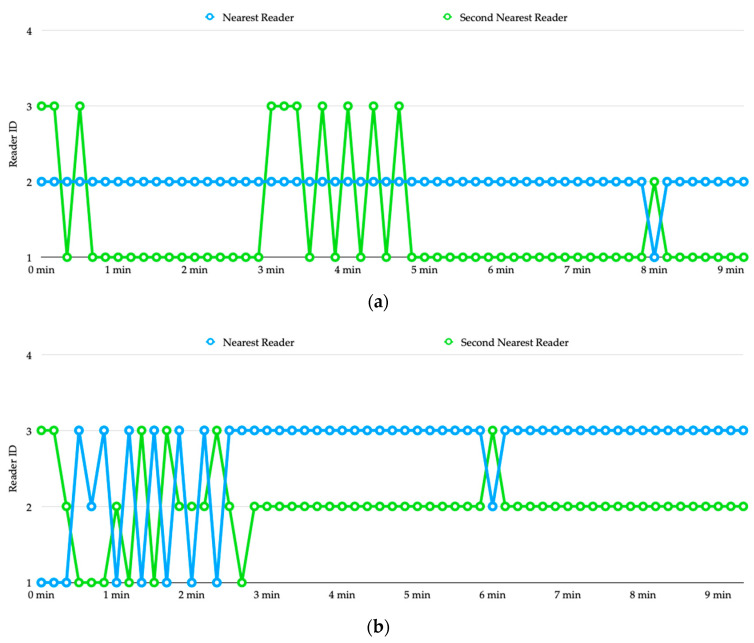
Experiment 2: (**a**) determination of occupancy for vehicle 1, (**b**) determination of occupancy for vehicle 2.

**Figure 13 sensors-20-04559-f013:**
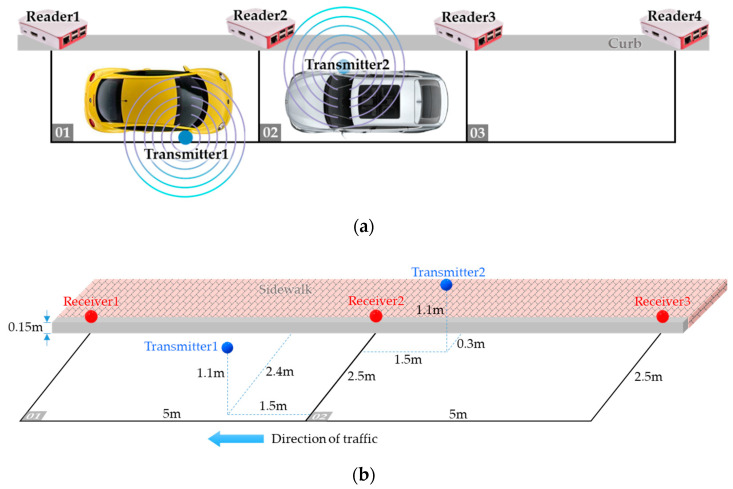
Experiment 3: (**a**) test scenario, (**b**) detailed deployment within parking space 01 and 02.

**Figure 14 sensors-20-04559-f014:**
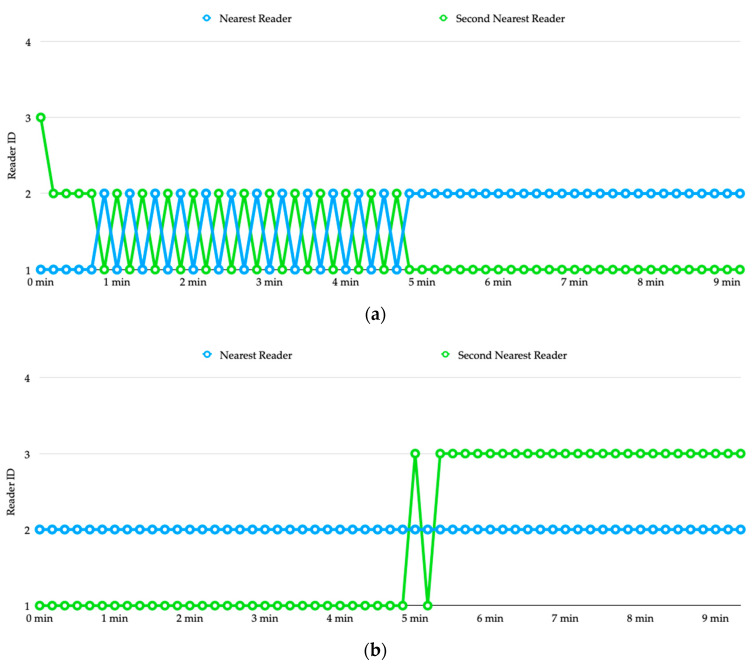
Experiment 3: (**a**) determination of occupancy for vehicle 1, (**b**) determination of occupancy for vehicle 2.

**Figure 15 sensors-20-04559-f015:**
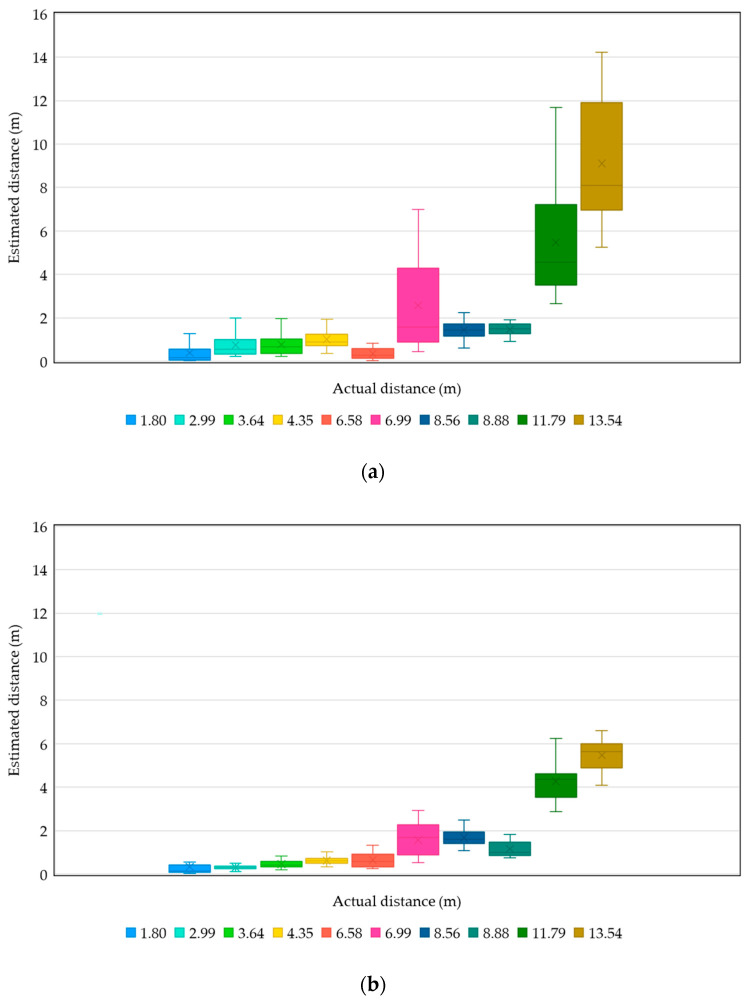
Boxplots of transmitter-to-receiver distance estimations: (**a**) data from the first 5 min (**b**) data from the second 5 min.

**Table 1 sensors-20-04559-t001:** Sample data collected by the gateway.

Reader ID	MAC Address of the Transmitter	Estimated Distance (m)	Time of Detection
1	11:22:33:44:55:66	7.45	2020-06-25 15:13:10
2	11:22:33:44:55:66	1.36	2020-06-25 15:13:08
3	11:22:33:44:55:66	4.03	2020-06-25 15:13:12
4	11:22:33:44:55:66	10.28	2020-06-25 15:13:11

**Table 2 sensors-20-04559-t002:** List of equipment in the experiments.

Item	Specification
Beacon transmitter	Estimote location beacon (Bluetooth 5.0)
Beacon reader	Raspberry Pi 3 Model B+
Gateway	Raspberry Pi 3 Model B
Parking server	Desktop PC with Windows 10

**Table 3 sensors-20-04559-t003:** Comparison of existing infrastructure-based on-street parking systems.

	Magnetometer-Based Systems	Smart Parking Meters-Based Systems	Mackey et al.’s Beacon-Based System	Our System
Automatic detection of parking space occupancy?	Yes	Yes	No	Yes
Automatic detection of vehicle ID?	No	Yes	No	Yes
Number of roadside devices	*N*	*N*	*N*	*N* + 1
Total cost of deployment of roadside devices (USD)	100 × N	2000 × N	5 × N	10 × (N + 1)
Human labor required?	Yes (parking attendants)	No	Yes (both drivers and parking attendants)	No

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
