# Peer review of "A Low-Cost On-Street Parking Management System Based on Bluetooth Beacons†"

_sensors, 2020, doi:10.3390/s20164559_

Round 1

Reviewer 1 Report

The authors propose and describe their methodology in this study, my comments are:

  1. I am very concerned about the discussion and conclusions, it appears that there is no discussion, please improve it.
  2. Improve the conclusions, specifying the methods that are compared to. 

Reviewer 2 Report

The paper brings good description of existing smart parking systems. The proposed parking management system should be described better with system architecture Fig. 2. Also figures 3,4,5,6 are not informative and should be removed from paper. Experiments should be described more detailed with list of measurement equipment. The paper should be added with result analysis and comparison charts.

The article should be added with economics results of using such Bluetooth system architecture. It will be interesting to know about key benefits of the implementation for private and government parking places.

Reviewer 3 Report

This article proposed a low-cost on-street parking management system based on Bluetooth beacons, it was a system including readers, transmitters, gateway and parking server. The system proposed was complete and could achieve the goal of detecting the occupancy and acquire the identity of vehicles, with lower cost than wireless management meters and image recognition-based smart parking meters. However, I have some questions and advises to your system:

  • The transmitters placed on cars should send beacon packets initiatively, what if the car is not install with such a transmitter, or it intentionally does not install an available transmitter, can it park or not? How to stop these cars parking without a matching transmitter?
  • In your implementation, you chose the iBeacon format for beacon packets, different formats are not compatible to each other, also, a special UUID was used for parking service. Since packet format and UUID make the transmitter and reader should match, all on-street parking places and vehicles need to install unite devices, which is a huge municipal engineering to government. Then, how to make your system universal?
  • What if one vehicle occupied two or more parking lots, how to detect this situation?
  • I advise to add an appeal entry point to the parking App, in case the record and payment bugs sometimes.

Round 2

Reviewer 1 Report

The authors followed all the recommendations, my decision is to accept the paper.